# Identification of Intrinsically Disordered Proteins and Regions in a Non-Model Insect Species *Ostrinia nubilalis* (Hbn.)

**DOI:** 10.3390/biom12040592

**Published:** 2022-04-18

**Authors:** Miloš Avramov, Éva Schád, Ágnes Révész, Lilla Turiák, Iva Uzelac, Ágnes Tantos, László Drahos, Željko D. Popović

**Affiliations:** 1Department of Biology and Ecology, Faculty of Sciences, University of Novi Sad, 21000 Novi Sad, Serbia; milos.avramov@dbe.uns.ac.rs (M.A.); iva.uzelac@dbe.uns.ac.rs (I.U.); 2Institute of Enzymology, Research Centre for Natural Sciences, 1117 Budapest, Hungary; schad.eva@gmail.com (É.S.); tantos.agnes@ttk.hu (Á.T.); 3Institute of Organic Chemistry, Research Centre for Natural Sciences, 1117 Budapest, Hungary; revesz.agnes@ttk.hu (Á.R.); turiak.lilla@ttk.hu (L.T.); drahos.laszlo@ttk.hu (L.D.)

**Keywords:** intrinsically disordered proteins (IDPs), intrinsically disordered protein regions (IDRs), LC–MS/MS, IUPred analysis, *Ostrinia nubilalis*, cold hardiness

## Abstract

Research in previous decades has shown that intrinsically disordered proteins (IDPs) and regions in proteins (IDRs) are as ubiquitous as highly ordered proteins. Despite this, research on IDPs and IDRs still has many gaps left to fill. Here, we present an approach that combines wet lab methods with bioinformatics tools to identify and analyze intrinsically disordered proteins in a non-model insect species that is cold-hardy. Due to their known resilience to the effects of extreme temperatures, these proteins likely play important roles in this insect’s adaptive mechanisms to sub-zero temperatures. The approach involves IDP enrichment by sample heating and double-digestion of proteins, followed by peptide and protein identification. Next, proteins are bioinformatically analyzed for disorder content, presence of long disordered regions, amino acid composition, and processes they are involved in. Finally, IDP detection is validated with an in-house 2D PAGE. In total, 608 unique proteins were identified, with 39 being mostly disordered, 100 partially disordered, 95 nearly ordered, and 374 ordered. One-third contain at least one long disordered segment. Functional information was available for only 90 proteins with intrinsic disorders out of 312 characterized proteins. Around half of the 90 proteins are cytoskeletal elements or involved in translational processes.

## 1. Introduction

Intrinsically disordered proteins exist and function without a well-defined three-dimensional structure, occupying a conformational space through a series of fluctuating structural states, often described as conformational ensembles [1].

Intrinsic disorder can span the full length of a polypeptide chain or be localized in specific regions of globular, ordered proteins as intrinsically disordered protein regions or IDRs [2]. A protein’s amino acid sequence contains the code for disorder [3], as certain residues were found to be particularly abundant in intrinsically disordered proteins (IDPs) and IDRs, such as Ala, Glu, Ser, Gln, Lys, and Pro [4].

The biological importance of intrinsic disorder is well described: many proteins that are involved in signaling and regulatory pathways, as well as different intermolecular interactions, possess segments that are unstructured. This allows them to recognize a wide array of binding partners, often undergoing disorder-to-order transition upon interaction with their targets [5,6]. Disorder is present in all domains of life, even in viruses. It is established that the total disorder content, measured in the amount of disordered residues present in the entire proteome, is higher in eukaryotes than in prokaryotes and viruses [7,8,9]. Despite the abundance of IDPs/IDRs in proteomes, as well as their key roles in regulatory processes, experimental characterization of these proteins is still sparse. The fact that IDPs and ID-containing proteins are difficult to crystalize makes them unsuitable for structure-resolving methods such as X-ray crystallography, while high protein disorder can prohibit structure determination using cryo-EM. Additionally, cryo-EM has a rather strict and high size restriction, so its usefulness in the structure determination of individual proteins is rather limited [10,11]. It has also been reported that nearly a half of eukaryotic proteins are considered parts of the “dark proteome”, meaning there is a lack of information on their folded structure which prevents them from being used in homology modeling [12], further complicating the annotation of IDPs and IDRs [13,14]. While experimental characterization has its share of obstacles, different computational methods have been developed that can be used to predict disorder in protein sequences as well as to facilitate functional annotation of such proteins *in silico* [15,16,17,18].

In this article, a combined approach to identifying intrinsically disordered proteins is presented. Wet lab methods are employed to generate protein samples and prepare them for bioinformatic analysis. A particular experimental setup provides biological context for the subsequent computational analysis of identified proteins. For example, intrinsically disordered proteins are known to possess cold stability and resistance to cold treatment [19], so they likely play important protective roles in the mechanisms behind acclimation and adaptation to cold winter temperatures. In order to study this aspect of IDPs, a pipeline for the identification of such proteins in non-cold-acclimated and cold-acclimated diapausing larvae of a moth insect species was developed. The insect in question, the European corn borer (ECB, *Ostrinia nubilalis*, Hbn.), is a Eurasian species of moth that was introduced to North Africa and North America in the early 1900s by accident [20]. Larvae of the ECB are notorious as pests that feed on more than 200 economically important species of grains, fruits, and vegetables [21,22]. The larvae used in this study were acclimated to temperatures below 0 °C, triggering specific molecular, biochemical, and behavioral adaptations in this species that should also lead to changes in the content of its proteome. This way, protein identification and characterization can be examined in the context of the cold hardiness molecular ecophysiology in this species. Thus, the broader aim of this study is to explore how intrinsic disorder is implicated in the adaptation of this species to cold weather conditions. In the months preceding winter, the fifth instar larvae of the ECB enter diapause, a form of life cycle arrest observed mostly in insects of temperate and polar zones, in order to prepare for the coming cold temperatures and food scarcity [23]. Diapause consists of three major phases—pre-diapause, diapause, and post-diapause, allowing an organism to gradually adapt to the changes in the environment and ensuring its survival [24]. An organism undergoes a wide array of molecular and biochemical adaptations during these different diapause phases, such as depression of metabolism [25,26], alteration of metabolic enzyme activity [27], synthesis of specific cryoprotectants [28,29], and changes in the lipid composition of storage molecules and membranes [30,31,32], as well as changes in the expression of genes and proteins involved in stress protection and the regulation of cell cycle and programmed cell death [33,34,35,36,37,38]. The listed adaptations allow the diapausing larvae of the ECB to develop cold hardiness and successfully overwinter [23,39].

An added value of this approach is its robustness; hence, it can be adjusted and used in research with other types of biological materials and experimental setups.

## 2. Materials and Methods

### 2.1. Experimental Design

The generalized experimental workflow is presented in Figure 1. Diapausing (winter) fifth instar larvae of the ECB were collected from the fields of the Maize Research Institute in Zemun Polje (44°87′ N, 20°33′ E), Serbia, during the winter season of 2018/2019. The collected larvae were first acclimated at 15 °C for two weeks, after which one group was frozen in liquid nitrogen and stored at −80 °C until analysis. The remainder of the larvae were placed in insect homes made out of waxed cardboard and gradually chilled by lowering the temperature by 1 °C each day, with additional acclimation for two weeks when specific checkpoint temperatures were reached (5, −3, and −16 °C). After the final acclimation, larvae were frozen in liquid nitrogen and stored at −80 °C until analysis. In total, two experimental groups were formed—one non-cold-acclimated (NCA) diapausing group and one cold-acclimated (CA) diapausing group (Figure 1). The groups consisted of 5 biological replicates each.

### 2.2. Sample Preparation

Whole-body larvae (5 per sample) were homogenized in ice-cold 50 mM K-phosphate buffer pH 7.5 with 1 mM DTE to make a 20% *w*/*v* homogenate. The homogenates were then additionally lysed with sonication for 2 min (24 sonic pulses lasting 5 s each, with 10 s pauses in-between). Sonication was followed by centrifugation for 10 min at 7000 rpm, 4 °C to remove insoluble debris and lipids. Supernatants were divided into two microtubes per sample. One microtube from each sample was placed in a water bath at 100 °C for 5 min in order to remove globular proteins and enrich the content of IDPs (heated sample type, Figure 1). The other microtube was left untreated for comparison (non-heated sample type, Figure 1). After the heat treatment, all samples were centrifuged for 10 min at 12,000 rpm, 4 °C to further purify them and the supernatants were transferred to clean microtubes. Total proteins were assayed on a 250 µL microplate using the commercial Quick Start™ Bradford Protein Assay kit (Bio-Rad, Hercules, CA, USA, cat. no. 5000203), according to the manufacturer’s protocol. After the assay, a 5× concentrated protease inhibitor cocktail (Roche cOmplete™ ULTRA tablets, Merck KGaA, Darmstadt, Germany, cat. no. 5892970001) was added to the samples to a final concentration of 1×.

### 2.3. Protein Identification

Protein identification was done using shotgun LC–MS/MS and the Mascot search engine [40]. Aliquots containing up to 20 µg of total protein were taken from every whole-body homogenate, and proteins were double-digested in-solution using Trypsin/Lys-C mixture, followed by Trypsin digestion (Figure 1). First, the samples were prepared for digestion in Microcon-10 kDa centrifugal filters according to the following steps:Rinse the filters with 200 µL of LC-MS grade H_2_O by centrifuging at 13,500× *g*, 4 °C for 10 min, with ~30 µL of water remaining in the filter after the rinse; discard the elute from the outer vial;Add a solution of up to 20 µg of protein to the filter and fill up to 200 µL with 200 mM NH_5_CO_3_, centrifuge at 13,500× *g*, 4 °C for 10 min; discard the eluate from the outer vial;Add 200 µL of 200 mM NH_4_CO_3_, centrifuge at 13,500× *g*, 4 °C for 10 min; discard the eluate from the outer vial;Add 200 µL of 50 mM NH_4_CO_3_, centrifuge at 13,500× *g*, 4 °C for 10 min; discard the eluate from the outer vial;Place the filter upside-down in a new outer vial and centrifuge at 1000× *g* for 2 min to transfer the protein solution from the filter to the outer vial, then pipette the solution into a 0.5 mL Lo-Bind Eppendorf tube.

After the preparation, the protein samples were digested according to the following protocol:Add 1.5 µL of LC-MS MeOH to the protein sample for a final MeOH concentration of 5%;Add 5 µL of 0.5% Rapigest and 2 µL of 200 mM DTT to the protein sample and incubate at 60 °C for 30 min;Cool the sample to room temperature, add 5 µL of 200 mM NH_5_CO_3_ and 2.5 µL of 200 mM iodoacetamide;Incubate in the dark for 30 min at room temperature;Add 1 µL of stock Trypsin/Lys-C Mix (20 µg of the mixture in 80 µL of LC–MS grade H_2_O) and incubate at 37 °C for 1 h;Add trypsin in a 1:25 trypsin:protein ratio at 37 °C for 1 h;Stop the digestion by adding 1.5 µL of formic acid for a final concentration of at least 2% *v/v*;Completely dry the samples in a vacuum dryer at 50 °C.

After the digestion, the samples were desalted and cleaned up using Pierce C 18 Spin Columns placed in 2 mL Lo-Bind Eppendorf tubes according to the following steps:Add 200 µL of 50% MeOH to the column and centrifuge at 1500 rpm for 1 min. Repeat the step once more and then discard the eluate;Add 200 µL of 0.5% TFA, 5% ACN solution to the column and centrifuge at 1500 rpm for 1 min. Repeat the step once more and then discard the eluate;Add 200 µL of 0.1% TFA to the column and centrifuge at 1500 rpm for 1 min. Repeat the step once more and then discard the eluate;Dissolve the dried sample in 50 µL of 0.1% TFA and apply it on the column. Centrifuge at 1500 rpm for 1 min;Collect the eluate and apply it on the column again. Centrifuge at 1500 rpm for 1 min;Add 100 µL of 0.1% TFA to the column and centrifuge at 1500 rpm for 1 min. Repeat the step once more;Place the column in a new 2 mL Lo-Bind Eppendorf tube;Add 50 µL of 0.1% TFA, 70% ACN solution to the column and centrifuge at 1500 rpm for 1 min to elute the sample. Repeat the step once more;Completely dry the sample in a vacuum dryer at 50 °C and store it in a freezer until analysis.

Tryptic digests were subjected to nano-LC–MS/MS analysis using a Dionex Ultimate 3000 RSLC nanoLC system (Sunnyvale, CA, USA) coupled to Bruker Maxis II ETD Q-TOF instrument (Bremen, Germany) via a CaptiveSpray nanoBooster ionization source. Peptides were separated online using Acquity M-Class BEH130 C18 analytical column (1.7 μm, 130 Å, 75 μm × 250 mm Waters, Milford, MA, USA) following trapping on an Acclaim PepMap 100 C-18 trap column (5 μm, 100 Å, 100 μm × 20 mm, Thermo Fisher Scientific, Waltham, MA, USA). The temperature was set at 48 °C, and a flow rate of 300 nl/min was applied. The gradient method was from 4% B to 50% B in 90 min; solvent A was 0.1% formic acid in water, while solvent B was 0.1% formic acid in acetonitrile.

Sample ionization was achieved in the positive electrospray ionization mode. Data-dependent analysis was performed using a fixed cycle time of 2.5 s. MS spectra were acquired over a mass range of 150–2200 *m*/*z* at 3 Hz, while CID was performed at 16 Hz for abundant precursors and at 4 Hz for ones of low abundance.

Data were evaluated with ProteinScape 3.0 software (Bruker Daltonic GmbH, Bremen, Germany) using the Mascot search engine version 2.5.1 (Matrix Science, London, UK). MS/MS spectra were searched against *O. nubilalis*, *O. furnacalis* (a species closely related to the ECB), as well as all lepidopteran protein sequences available in the NCBI database, due to the limited availability of *O. nubilalis* protein sequences. The following parameters were applied: trypsin as enzyme, 7 ppm peptide mass tolerance, 0.05 Da fragment mass tolerance, and 2 missed cleavages. Carbamidomethylation on cysteines was set as a fixed modification, with deamidation (NQ) and oxidation (M) as variable modifications.

### 2.4. 2D PAGE

To validate the results of IDP detection, proteins from both untreated and heat-treated samples were separated using a modified in-house 2D PAGE method [41]. Aliquots from the treated and untreated samples were taken and pooled in two mixtures, respectively (Figure 1). In the first dimension, the mixtures were run on a discontinuous native PAGE (12.5% separating gel, 20 µg of total proteins per well) for 50 min at 180 V. After the run, individual lanes were cut out as strips and placed in 1.5 M Tris-HCl pH 8.8 containing 8 M urea for 45 min to solubilize the proteins. The separating gel (12.5%) for the second dimension was prepared by adding 8 M urea to the standard native gel solution. After casting the gel, a strip with solubilized proteins was placed on top of it instead of a stacking gel, making sure not to introduce any bubbles between the separating gel and the strip. The second dimension was run for 30 min at 400 V. After the run, the gels were stained using the Pierce™ Silver Stain Kit (Thermo Scientific, Waltham, MA, USA, cat. no. 24612) to visualize the protein spots.

### 2.5. Bioinformatic Analysis

After protein identification, FASTA sequences for all identified proteins were downloaded from the NCBI database to be used for the prediction of structural disorder (Figure 1). The structural disorder of proteins was determined with the IUPred long disorder predictor (https://iupred2a.elte.hu/, accessed on 24 March 2022) [42], which is based on estimating the total pair-wise inter-residue interaction energy gained upon folding of a polypeptide chain. An amino acid is considered to be disordered if its IUPred score is at least 0.5. Mean disorder was computed as the average of residue scores, which range from 0.0 to 1.0. Overall disorder rate (percental disorder, ranging from 0 to 100%) represents the fraction of disordered amino acids in a polypeptide chain. Proteins are considered globular if their overall disorder rate is below 10%; nearly ordered if the rate is between 10% and 30%; partially disordered if the rate is between 30% and 70%; (mostly) disordered if the rate is above 70%. All proteins were further analyzed for the presence of long intrinsically disordered regions (long IDRs)—sequences of at least 20 consecutive disordered amino acids. Additionally, the amino acid composition of the proteins was analyzed to determine the absolute number of individual amino acids that make up each polypeptide, as well as their ratios.

Lastly, functional characterization was performed on the identified sequences (Figure 1). Functional information on the identified proteins was collected from various databases such as UniProt (www.uniprot.org/, accessed on 18 January 2022) [43], Pfam (http://pfam.xfam.org/, accessed on 18 January 2022) [44], Interpro (https://www.ebi.ac.uk/interpro/, accessed on 18 January 2022) [45], and GeneOntology (http://geneontology.org/, accessed on 18 January 2022) [46,47]. Data on their molecular functions, cellular localization, the biological processes they are involved in, and the domains they contained was collected. All of the analyses were performed using homemade PERL scripts run locally.

## 3. Results

Protein identification was performed directly from the individual homogenates, as described in the Methods section. In total, 820 proteins were identified—506 in the non-cold-acclimated (NCA) group and 314 in the cold-acclimated (CA) group. Accounting for shared entries between the two experimental groups, our investigation yielded a total of 608 unique proteins (Appendix A), with almost 50% of hits (290) being linked to polypeptides that have only been predicted from nucleotide sequences. Out of that total, 294 were present only in the NCA experimental group, with 102 only in the CA experimental group; the remaining 212 proteins were found in both (Figure 2A).

Identification of proteins from complex mixtures, such as larval extracts used in this study, can be challenging for a number of reasons. The signal of less abundant proteins can be masked by the ones that are overrepresented in the samples, and proteins embedded in large, multi-subunit complexes may remain invisible to LC–MS/MS. Heat treatment of such samples can enable the detection of those proteins, with the added advantage of enriching proteins with significant disorder content. A comparison of total identified proteins was made between the heat-treated and untreated samples of both experimental groups (Figure 2B). Heating the samples resulted in the identification of an additional 180 unique proteins, compared to the non-heated sample in the NCA group, while 265 heat-sensitive proteins were eliminated. The two sample types had 61 proteins in common. Within the CA group, 96 proteins were found only in the heated sample, 184 in the non-heated samples, and 34 proteins were shared between the two sample types.

To assess the disorder content of the identified proteins, a reliable disorder predictor, IUPred, was used to determine the disorder tendencies of the hits (Appendix A). Percental disorder was calculated for both the non-cold-acclimated and cold-acclimated diapausing groups (506 and 314 proteins, respectively). Proteins with an average percental disorder of 70% or higher were considered as mostly disordered (MDPs), with partially disordered proteins (PDPs) if the average percental disorder was between 30% and 70%, nearly ordered (NOPs) for values between 10% and 30%, and ordered (OPs) if the percental value was no higher than 10%. In the NCA group, MDPs accounted for 31 of all identified proteins; 81 were PDPs, 75 were NOPs, and the remaining 319 were OPs. In the CA group, 16 proteins were MDPs, 51 were PDPs, and 45 were NOPs; there were 198 OPs (Figure 3A). The heat treatment had a remarkable effect on the distribution of proteins, with various degrees of intrinsic disorder in both experimental groups. The heat-treated samples contained more partially and mostly disordered proteins compared to the non-heated samples, while still retaining a significant portion of heat-resistant ordered proteins (Figure 3B).

Since percental disorder in itself is not necessarily informative of function, it is better to identify long intrinsically disordered regions (long IDRs), which have a higher potential to possess biological relevance. A region is considered a long IDR if it contains at least 20 consecutive disordered amino acids. Further analysis was performed to determine whether the proteins identified in this study contain long IDRs (Appendix A). The results show that the proteins from both the NCA (Figure 4A) and CA (Figure 4B) experimental groups follow mostly similar patterns when it comes to the distribution of long IDRs in their sequences. Ordered proteins are devoid of IDRs, as are most of the NOPs. The majority of PDPs contain either 1 or 2 such regions. Certain muscle proteins, on the other hand, such as myosin heavy chain, are particularly enriched in long disordered regions. Depending on the isoform, they possess between 6 and as many as 14 such segments in their sequence. When it comes to MDPs, most of them possess one disordered segment, followed by proteins containing two, four, and three long IDRs, respectively.

The amino acid composition of every identified protein was analyzed, and the ratios of individual amino acids that make up the polypeptides were determined (Figure 5). Depending on the degree of structural disorder, the proteins in this dataset are comprised of varying amounts of disorder- and order-promoting amino acids. On average, mostly disordered proteins are composed of 68.7% disorder-promoting and 31.3% order-promoting amino acids. Partially disordered proteins have a similar composition to MDPs (65.2%/34.8%), while nearly ordered and ordered proteins trend towards a more balanced distribution of disorder- and order-promoting amino acids—60.5%/39.5% and 56.1%/43.9%, respectively.

The MDPs and PDPs identified in this study are particularly enriched in disorder-promoting amino acids (Figure 6A), such as glutamate (12.07% and 11.77% of total amino acids in a sequence on average, respectively), lysine (9.63% and 8.54%), and glutamine (6.69% and 5.34%), compared to nearly ordered (7.69% E, 8.39% K, and 4.28% Q) and ordered proteins (6.54% E, 7.2% K, and 3.48% Q). The only exception is glycine, which is more prevalent in nearly ordered (7.41%) and ordered proteins (7.46%) than in MDPs and PDPs (5.95% and 5.85%, respectively). Additionally, MDPs contain almost double the amount of proline as the other protein groups. When it comes to order-promoting amino acids (Figure 6B), MDPs are almost depleted in cysteine (0.39%), tyrosine (1.86%), phenylalanine (1.88%), isoleucine (3.67%), and valine (5.39%) in comparison to nearly ordered (0.98% C, 2.49% Y, 3.48% F, 5.23% I, and 7.26% V) and ordered proteins (1.88% C, 3.3% Y, 4.04% F, 5.78% I, and 7.43% V). Leucine is the standout order-promoting amino acid that partially disordered proteins contain in amounts similar to NOPs and OPs (7.88% compared to 7.54% and 8.58%, respectively), unlike MDPs (5.53%). The distribution of the remaining amino acids is more or less similar between the three groups of proteins.

*In silico* structure predictions should ideally be supported by detailed *in vitro* structural studies, which are difficult to carry out on a proteomic scale. In order to validate the computational analyses and identification of intrinsically disordered proteins, a specific two-dimensional electrophoretic assay [41] was performed. The assay can provide experimental information on the large-scale structural state of a protein solution. It is based on the heat-stability and resistance to chemical denaturation of IDPs, resulting in a pattern where disordered proteins align in a diagonal line in the second dimension. For this step, two 2D PAGE were performed, one for each sample type (untreated and heat-treated). In order to ensure that as many unique proteins were covered by the 2D PAGE, aliquots from both experimental groups (NCA and CA) were pooled and run as a singular sample on the respective gels. As seen in Figure 7A, a large proportion of the proteins from the heat-treated sample are aligned on the diagonal line, signifying that they are mostly or fully disordered. The heat-stable globular proteins are generally found above the diagonal line in this setup, as indicated by the arrows in Figure 7A. Additionally, it can be seen that proteins from the non-heated sample, which is rich in globular polypeptides, mostly stayed in the first gel and did not transfer into the second gel. In fact, the proteins did not migrate far during the separation in the first dimension (Figure 7B). This is likely due to the abundance of high molecular weight arylphorins, common storage proteins in the hemolymph of the ECB [48], which prevented other proteins from separating. Heating of the samples and removal of globular proteins resolved this issue, which allowed the proteins to separate in the first dimension and transfer into the second gel.

To gain an insight into the biological importance of IDPs in the cold adaptation process of the ECB, we performed a bioinformatic functional analysis of the identified disordered proteins using the data from online knowledgebases Uniprot, Pfam, Interpro, and Gene Ontology. Our results have revealed that only 312 of the proteins have a Uniprot entry and at least one data point from the other listed knowledgebases. Out of that number, 90 proteins are either mostly or partially disordered or contain at least 1 long IDR (Figure 8, Total unique).

The largest functional group (24 unique hits) comprises the structural components of the cytoskeleton or proteins associated with it, such as actin filament organization proteins or regulators of muscle contraction. The second-largest group comprises proteins functioning as molecular chaperones (21 unique hits), followed by proteins involved in translational processes (10 unique hits). The rest of the proteins cover a wide range of biological processes and molecular functions, such as protein and amino acid metabolism (9 unique hits), binding of nucleic acids (6 unique hits), binding of chitin and cuticle formation (4 unique hits), and others (Figure 8, Total unique). Additionally, heat treatment of the samples increased the number of proteins that could be identified and functionally analyzed. Proteins that are involved in chitin-binding and cuticle formation were found only in the heated samples, as were the majority of nucleic-acid-binding proteins (5 total hits compared to 1 total hit). More proteins belonging to the Cytoskeleton category were also present in the heated samples (19 total hits) compared to the non-heated ones (10 total hits). More proteins that act as molecular chaperones, on the other hand, were present in the non-heated samples (16 total hits) than in the heated samples (6 total hits) (Figure 8, Non-heated, Heated).

## 4. Discussion

Proteomic identification of intrinsically disordered proteins is still a field where improvements are needed. Here, we applied an effort where enrichment based on heat treatment and bioinformatics analysis were combined to identify as many IDPs as possible that are involved in the cold adaptation of a non-model insect species *O. nubilalis*. 

Almost three times as many unique proteins were identified in the NCA group than in the CA group (294 and 102, respectively, Figure 2A). The discrepancy is likely due to the experimental design, as cold acclimation in this species leads to a general depression of metabolic rate [25,26] and redirection of metabolic pathways that are of low priority towards the synthesis of low-molecular-weight cryoprotective compounds [27,29,49], among other things. When it comes to the effects of sample heating on the total number of proteins identified, as a means of IDP enrichment, fewer proteins were identified in the heat-treated samples, as expected. However, heat treatment enabled the identification of many proteins that were masked from LC–MS/MS analysis in the non-treated samples (Figure 2B). Additionally, it is important to highlight that nearly 50% of the proteins identified in this study have previously only been predicted from nucleotide sequences and were thus experimentally validated.

Structural disorder was predicted for the identified proteins using the IUPred algorithm. Depending on the amount of intrinsic disorder, the proteins were divided into four categories—mostly disordered proteins (MDPs, at least 70% percental disorder), partially disordered proteins (PDPs, 30–70% percental disorder), nearly ordered proteins (NOPs, 10–30% percental disorder), and ordered proteins (OPs, 10% percental disorder at most). According to our results, 30% of all identified proteins are either partially or mostly disordered or belong to the group of nearly ordered proteins that contain long IDRs (Figure 3A, Figure 4), in accordance with previous meta-studies on the prevalence of protein intrinsic disorder in eukaryotic proteomes [7,8,9,50]. The majority of these proteins were revealed after IDP enrichment by sample heating, as only a few of them were identified specifically in the non-heated samples (Figure 3B). Heating the samples led to the removal of globular and heat-sensitive disordered proteins, while the majority of disordered proteins remained unaffected due to their stability in denaturing conditions, such as high temperatures [51,52,53]. As such, the heat treatment had a remarkable effect on the distribution of proteins with various degrees of intrinsic disorder in both NCA and CA experimental groups. The heat-treated samples contained more partially and mostly disordered proteins compared to the non-treated samples while still retaining a significant portion of heat-resistant ordered proteins. These findings further underscore the importance of sample boiling, without which we would have not only identified fewer novel proteins, but also missed out on the disordered proteins. The remaining 374 proteins (61% of total) scored 10% percental disorder at most and were labeled as ordered globular proteins. The reason that most of these proteins have percental values above 0%, which would indicate a complete absence of intrinsic disorder, and are still considered ordered is that fully structured proteins are quite rare. In fact, only around 7% of proteins deposited in the Protein Data Bank (PDB) do not contain any disordered residues at all [54].

Adding on to the fact that even globular proteins can contain unstructured segments, all proteins were analyzed for the presence of such long intrinsically disordered regions (long IDRs). These are segments of at least 20 consecutive disorder-promoting amino acids. They can be associated with sites of post-translational modification, act as flexible linkers to facilitate domain movements, or function as regions of molecular recognition and binding [55,56]. Proteins that scored below 70% (PDPs, NOPs, and OPs, 569 in total) were analyzed for the presence of IDRs first. Around 75% of these proteins (427 in total) did not contain any long disordered regions. Most of them were the ones labeled as ordered proteins (355 out of 374 OPs) and nearly ordered proteins (52 out of 95 NOPs). However, a fifth of partially disordered proteins did not contain any long IDRs either (20 out of 100 PDPs). This discrepancy between the measured percental disorder and the presence/absence of long IDRs in these groups of proteins is probably indicative of the localization of disorder in their structures. Partially disordered proteins without long IDRs likely contain disordered residues in contiguous segments that do not reach the 20 amino acids threshold of long IDRs, yet the segments are long enough to be reflected on the measured IUPred score of at least 10%. When it comes to ordered proteins, the small amount of detected disorder that they do possess is likely contained within the singular long IDR that some of these proteins contain. In total, 142 proteins that scored below 70% percental disorder contain at least one long IDR, with most of them (55%) containing exactly one. More than a fifth of them contain two LDRs, while the remaining proteins contain at least 3 and up to 14 long disordered regions. Next, mostly disordered proteins were subjected to the same analysis. However, this was done in order to further characterize the nature of their structural disorder. The majority of these proteins (38%) contain one long disordered region in their sequences, but two and three such segments are also frequent. A small number of these proteins contain four or more long IDRs. Interestingly, it can be seen that around 20% of the identified proteins contain none, despite their high disorder content (Figure 4A,B).

An interesting approach is to also analyze the amino acid composition of the identified proteins. Considering that intrinsic disorder is encoded in the primary sequence of proteins [3], this type of analysis would provide valuable information on the distribution of individual amino acids in the varying degrees of structural disorder. Depending on their intrinsic properties, amino acids can promote either order or disorder in protein structure in varying degrees. In that sense, amino acids can be designated as order- (W, F, Y, I, M, L, V, N, C, T) or disorder-promoting (A, G, R, D, H, Q, K, S, E, P), with tryptophan being the most order-promoting and proline the most disorder-promoting amino acid [57]. On average, more than two-thirds of amino acids in the primary sequence of MDPs and PDPs identified in this study are disorder-promoting (Figure 5). In particular, these proteins are enriched in polar charged amino acids such as glutamic acid (E, 12.07% and 11.77%, respectively), lysine (K, 9.63% and 9.19%), and polar non-charged glutamine (Q, 6.69% and 5.34%) in comparison to NOPs and OPs that have noticeably lower amounts of these amino acids in their sequences (Figure 6A). Another amino acid that MDPs are conspicuously enriched in compared to the other groups of proteins is proline (7.52%, Figure 6A), which is known to disrupt the formation of secondary structures in protein [58,59]. On the other hand, it can be noticed that PDPs contain a considerable amount of leucine (L, 7.88%, Figure 6B), which is considered an order-promoting amino acid. Compositional analysis has shown that disorder-containing muscle proteins, such as tropomyosins, are particularly rich in leucine (10% or higher content), indicating the importance of this amino acid for the function of muscle proteins. The aforementioned tropomyosins, for example, are involved in the regulation of muscle contraction and contain leucine zippers in their C-termini [60], similar to many nucleic-acid-binding proteins such as transcription factors. Functional analysis shows that the identified proteins are involved in a wide array of biological processes and, as such, fulfill many different functions. The majority of the proteins (24) are in some way connected with the organization and operation of the cytoskeletal network. Either they are structural constituents of the cytoskeleton or they regulate the contraction of muscle fibers. Next, a faction of the disorder-containing proteins (21) was found to act as molecular chaperones and assist in the proper folding of nascent proteins or handling of misfolded polypeptides. The last major group of proteins (10) is the proteins that are involved in translational processes, such as translation elongation, or are structural components of ribosomes. The remaining proteins cover a myriad of functions and processes, such as the metabolism of carbohydrates, lipids, and proteins, formation of insect cuticle, binding of nucleic acids, synthesis of amino acids, and oxidoreductive processes in connection with the electron transport chain (Figure 8). Sample heating also led to the identification of novel functions/processes, such as proteins involved in the formation of insect cuticle. These proteins were found exclusively in the heated samples. Likewise, the fraction of proteins involved in the cytoskeletal network was enriched after sample heating, as were the proteins involved in the binding of nucleic acids (Figure 8, Heated). On the other hand, heat treatment led to the removal of many molecular chaperones from the affected samples, so this group of proteins was more prominent in the non-heated samples (Figure 8, Non-heated). These results lend even more credence to the importance of heat treating the samples when it comes to the identification and functional analysis of intrinsically disordered proteins.

The findings in this study, first and foremost, demonstrate the need for further, more thorough research into the identification and characterization of intrinsically disordered proteins and intrinsically disordered protein regions. Secondly, by combining wet and dry lab methods, as proposed here, valuable information on the pervasiveness and function of IDPs/IDRs can be uncovered. Even a simple experimental design, such as the acclimation of insect larvae to low temperatures, caused an evident differentiation in the proteome content between the two experimental groups, both in quantity and quality. This is a reflection of the different metabolic states these experimental groups are in, as well as the various changes that have occurred at the molecular and biochemical levels. As such, the roles and functions of the identified proteins can be inferred from this differentiation, even if actual functional data is missing from the relevant databases. Additionally, this gives direction on where next to take the research and what proteins to focus on.

## Figures and Tables

**Figure 1 biomolecules-12-00592-f001:**
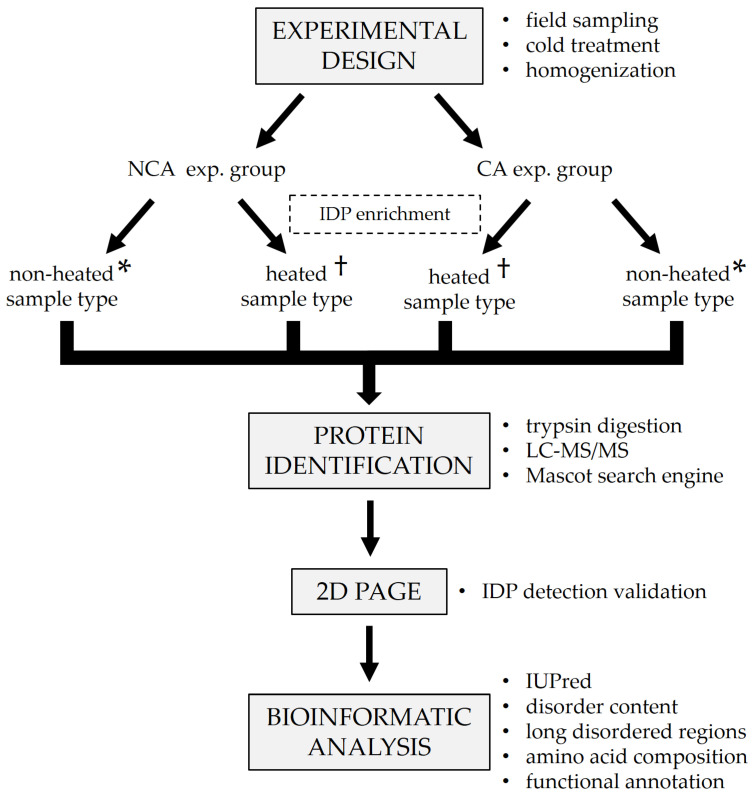
Generalized workflow of the experiment. Detailed explanations are given in the relevant subsections. Aliquots from samples with the same symbol in the superscript (* or †) were pooled before being run on 2D PAGE. NCA—non-cold-acclimated diapausing group; CA—cold-acclimated diapausing group.

**Figure 2 biomolecules-12-00592-f002:**
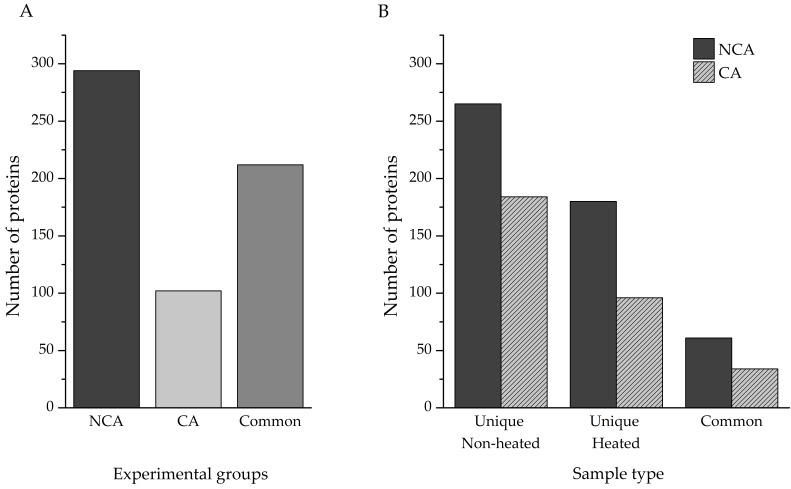
(**A**) Total unique and common proteins isolated from different experimental groups. (**B**) Effect of sample heating on the number of identified proteins. Unique Non-heated—proteins found only in the non-heated samples; Unique Heated—proteins found only in heated samples; Common—proteins that were found in both heated and non-heated samples; NCA—non-cold-acclimated diapausing group; CA—cold-acclimated diapausing group.

**Figure 3 biomolecules-12-00592-f003:**
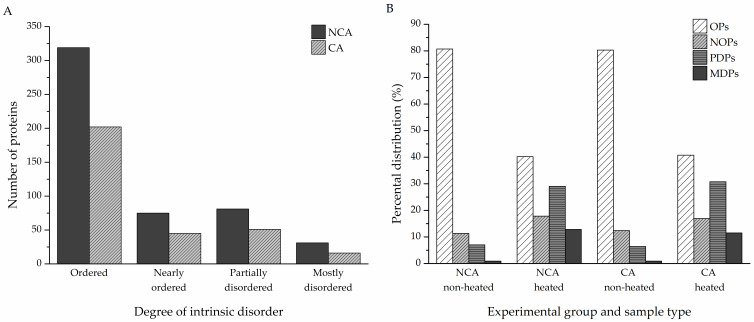
(**A**) Total number of proteins with varying degrees of intrinsic disorder—ordered (OPs, 10% at most), nearly ordered (NOPs, 10–30%), partially disordered (PDPs, 30–70%), mostly disordered (MDPs, at least 70%). (**B**) Effect of heat treatment on the percental distribution of proteins with varying degrees of intrinsic disorder in the two sample types of both experimental groups. NCA—non-cold-acclimated diapausing group; CA—cold-acclimated diapausing group.

**Figure 4 biomolecules-12-00592-f004:**
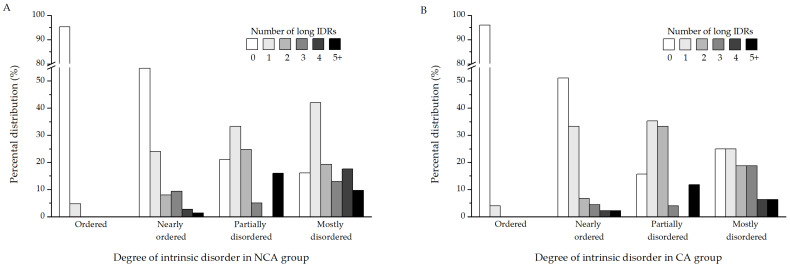
Distribution of long intrinsically disordered regions in proteins with various degrees of intrinsic disorder in the (**A**) NCA (non-cold-acclimated diapausing) and (**B**) CA (cold-acclimated diapausing) experimental groups. The different numbers of long IDRs in proteins are color-coded.

**Figure 5 biomolecules-12-00592-f005:**
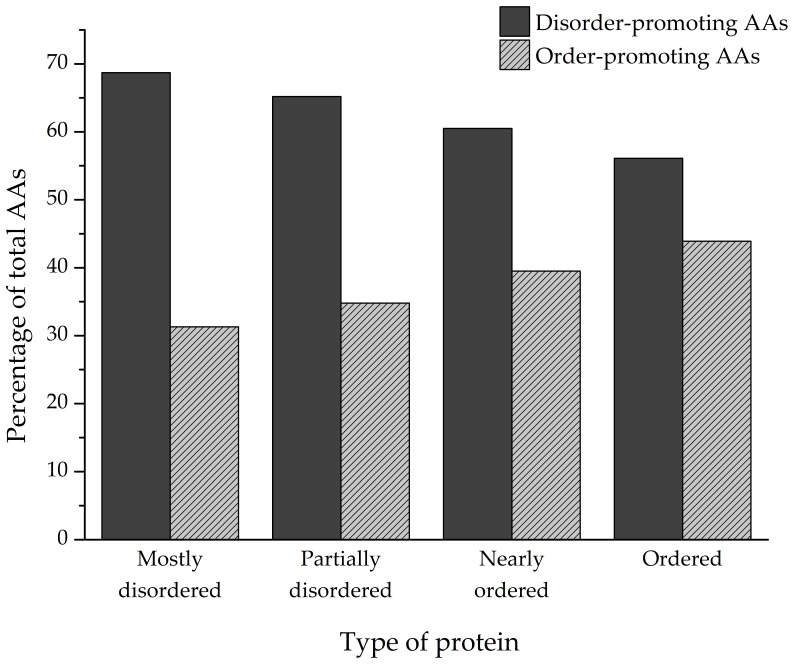
Ratios of disorder-promoting (P, E, S, K, Q, H, D, R, G, A) and order-promoting amino acids (T, C, N, V, L, M, I, Y, F, W) in mostly disordered (MDPs), partially disordered (PDPs), nearly ordered (NOPs), and ordered proteins (OPs).

**Figure 6 biomolecules-12-00592-f006:**
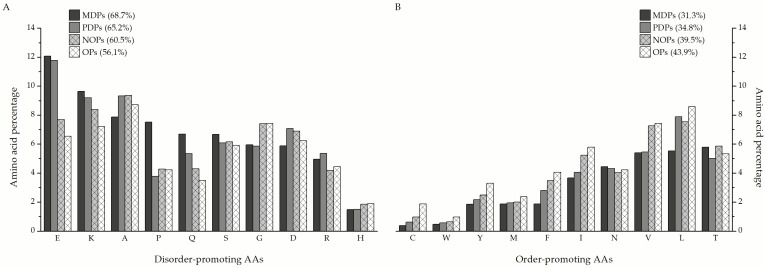
Ratios of individual disorder (**A**) and order-promoting amino acids (**B**) in mostly disordered (MDPs), partially disordered (PDPs), nearly ordered (NOPs), and ordered proteins (OPs), ordered by abundance in MDPs.

**Figure 7 biomolecules-12-00592-f007:**
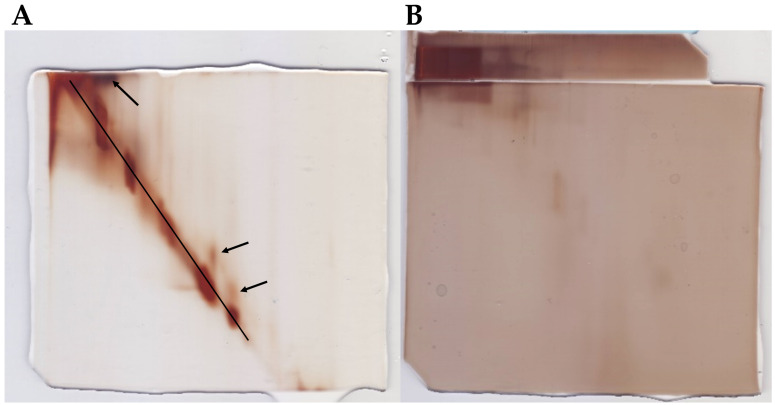
Results of in-house 2D PAGE for detecting intrinsically disordered proteins. (**A**) Proteins from heat-treated samples have successfully entered the second dimension. The black line represents the diagonal along which IDPs are located. Arrows denote ordered proteins that stay above the diagonal. (**B**) Proteins from non-heated samples are mostly locked in the gel from the first dimension (strip overlaying the larger gel).

**Figure 8 biomolecules-12-00592-f008:**
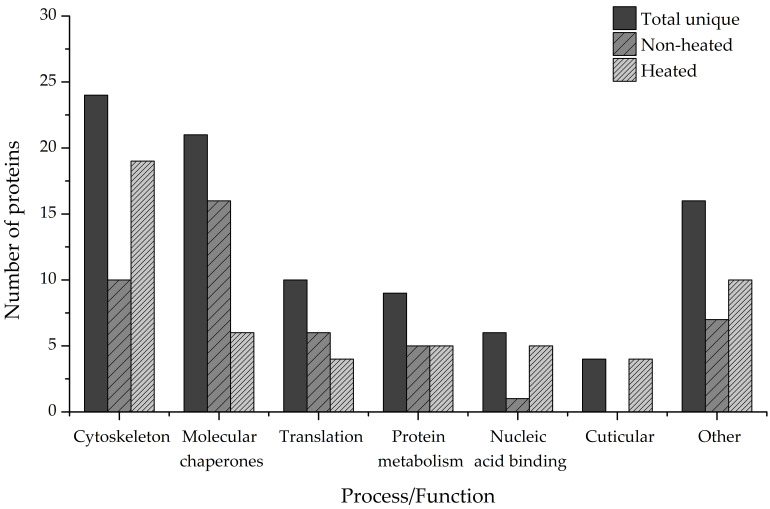
Biological processes and molecular functions of intrinsically disordered proteins and proteins containing long IDRs. The category Other encompasses processes and functions that make up less than 4% of total hits each. Total unique—all uniquely identified proteins; Non-heated—all proteins identified in the non-heated sample types; Heated—all proteins identified in the heated sample types.

## Data Availability

The data presented in this study are available from the corresponding author upon reasonable request.

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
