# Peer review of "Identification of Intrinsically Disordered Proteins and Regions in a Non-Model Insect Species Ostrinia nubilalis (Hbn.)"

_biomolecules, 2022, doi:10.3390/biom12040592_

Round 1

Reviewer 1 Report

The paper “Identification of intrinsically disordered proteins and regions 2 in a non-model insect species Ostrinia nubilalis (Hbn.)” by Avramov et al. explores how intrinsic disorder is implicated in the adaptation of this species to cold weather conditions.

In my opinion, the paper is of some interest, but it must be improved in order to become suitable for publication.

In particular: 

In predicting disordered regions, was the full a.a. sequence considered? My question arises because signal sequences, in general present at the N-terminus (for example, export signals), are always disordered. They should be excluded from the analysis, since these peptides are proteolyzed after export or secretion and they are not part of the mature protein.

The choice of the large central interval, 10%-70%, is quite questionable. This interval is quite large, 60% means that a large part of the protein is disordered, whilst 20%-30% disorder means that the protein is essentially ordered, with a relatively limited disordered region. It should be interesting to know if the disordered regions are evenly distributed in this large group 10%-70%. Or, better, to divide it in two groups, for example 30%-70% and 10%-30%. The latter group can be considered as that of proteins nearly-ordered (or proteins with a minor content of disorder). How many proteins belongs to each of the two groups? Does this change the number of proteins with IDRs?

In Fig 3A and B, percentages more than absolute numbers should be reported. For NCA group, 6.6%, 32.6% and 60.7% are MDP, PDP and OP proteins, respectively. For CA numbers are 5.3%, 32.1%, 62.4%. This means that the ratio between ordered and disordered proteins does not change, it is only the total number of proteins extracted (or produced by the organisms) that is different.

The most informative data are about IDRs (Lines 285-297). It should be interesting to know how many of these IDRs are at the N- or C-terminus and, eventually, exclude them from the statistics. This because a disordered peptide at the protein extremities is relatively common and in general it does not influence significantly protein function, whilst a disordered (or flexible) region inside the polypeptide sequence suggest the presence of two (or more) domains that can assume different relative orientations.

The experimental detection of IDP (electrophoresis part, Lines 329-345) is short and its relevance with respect to the rest of the paper is not evident. Is it there only to demonstrate that some proteins are disordered? It should be expanded or deleted.

Line 47-49 … X-ray crystallography and cryo-EM, … this statement is true for X-ray diffraction, since proteins must be crystallised, but not for cryo-EM, which uses single particles. The structure of proteins with a partial disorder can be determined using cryoEM, whilst this is nog possible only if the protein is fully or largely disordered.

Author Response

The authors would like to thank the Reviewer for the critique and suggestions regarding the manuscript. They have helped us to improve and enrich the article. We hope that the points have been adequately addressed. Please find below the response to the reviewer:

Reviewer comments:

The paper “Identification of intrinsically disordered proteins and regions 2 in a non-model insect species Ostrinia nubilalis (Hbn.)” by Avramov et al. explores how intrinsic disorder is implicated in the adaptation of this species to cold weather conditions.

In my opinion, the paper is of some interest, but it must be improved in order to become suitable for publication.

In particular: 

In predicting disordered regions, was the full a.a. sequence considered? My question arises because signal sequences, in general present at the N-terminus (for example, export signals), are always disordered. They should be excluded from the analysis, since these peptides are proteolyzed after export or secretion and they are not part of the mature protein.

Full a.a. sequence was considered for predictions. In that sense, the analysis was repeated and the termini were excluded. The manuscript body and figures have been updated to reflect this change.

The choice of the large central interval, 10%-70%, is quite questionable. This interval is quite large, 60% means that a large part of the protein is disordered, whilst 20%-30% disorder means that the protein is essentially ordered, with a relatively limited disordered region. It should be interesting to know if the disordered regions are evenly distributed in this large group 10%-70%. Or, better, to divide it in two groups, for example 30%-70% and 10%-30%. The latter group can be considered as that of proteins nearly-ordered (or proteins with a minor content of disorder). How many proteins belongs to each of the two groups? Does this change the number of proteins with IDRs?

The large central interval has been divided into the two suggested groups. In general, around half of the proteins were placed into the newly formed “nearly ordered” group, while the other half remained in the PDP group. The number of proteins with IDRs was mostly affected by the decision to exclude termini from the analysis.

The most informative data are about IDRs (Lines 285-297). It should be interesting to know how many of these IDRs are at the N- or C-terminus and, eventually, exclude them from the statistics. This because a disordered peptide at the protein extremities is relatively common and in general it does not influence significantly protein function, whilst a disordered (or flexible) region inside the polypeptide sequence suggest the presence of two (or more) domains that can assume different relative orientations.

This has been addressed.

In Fig 3A and B, percentages more than absolute numbers should be reported. For NCA group, 6.6%, 32.6% and 60.7% are MDP, PDP and OP proteins, respectively. For CA numbers are 5.3%, 32.1%, 62.4%. This means that the ratio between ordered and disordered proteins does not change, it is only the total number of proteins extracted (or produced by the organisms) that is different.

We feel that it is important to report total numbers in Figure 3A, as that highlights the differences in the total number of extracted proteins, caused by the cold acclimation treatment. On the other hand, Figure 3B has been updated to report percentages, as per suggestion. The differences in the distribution of proteins with different degrees of intrinsic disorder caused by IDP enrichment is now more evident.

The experimental detection of IDP (electrophoresis part, Lines 329-345) is short and its relevance with respect to the rest of the paper is not evident. Is it there only to demonstrate that some proteins are disordered? It should be expanded or deleted.

The electrophoresis further corroborates the necessity of sample heating in order to enrich the samples for IDPs. From the images it can be seen that sample heating has led to the removal of globular proteins which made the IDPs more visible to LC-MS/MS. This part has been expanded to include more detail.

Line 47-49 … X-ray crystallography and cryo-EM, … this statement is true for X-ray diffraction, since proteins must be crystallised, but not for cryo-EM, which uses single particles. The structure of proteins with a partial disorder can be determined using cryoEM, whilst this is nog possible only if the protein is fully or largely disordered.

The text has been updated for clarification.

Reviewer 2 Report

The manuscript submitted by Z. D. Popovic and coworkers compares the IDP and IDR content in proteins of cold acclimated and non-acclimated O. nubialis larvae.

The manuscript is repetitive. Many things described in the Results section are described again in the Discussion section.

Since I am not an expert in wet-lab procedures, I can comment only the bioinfo part, which is very simple: disorder predictions – only one method, IUPRED, was used – amino acid composition, collection of DB annotations.

However, the manuscript can be interesting for those interested in more realistic proteome analyses that include further variables – for example, temperature.

Line 24: “specific domains” might become “specific regions”.

Lines 57-58: The sentence “wet lab … tools” should be rewritten. It its present form, it makes no sense.

Section 2.1: perhaps one might write how many larvae have been used and how many of them were exposed to low temperature.

Line 237: perhaps “self-made” might become “home-made”.

Lines 481-482: the sentence “In silico structure predictions … structural studies,” does not make sense and should be rewritten. At this regard, the Authors should consider alphaFold2 and the related database developed at the EMBL.

Author Response

The authors would like to thank the Reviewer for the critique and suggestions regarding the manuscript. They have helped us to improve and enrich the article. We hope that the points have been adequately addressed. Please find below the response to the reviewer:

Reviewer comments:

The manuscript submitted by Z. D. Popovic and coworkers compares the IDP and IDR content in proteins of cold acclimated and non-acclimated O. nubialis larvae.

The manuscript is repetitive. Many things described in the Results section are described again in the Discussion section.

We have addressed this suggestion and redundant parts of the manuscript have been shortened our removed.

Since I am not an expert in wet-lab procedures, I can comment only the bioinfo part, which is very simple: disorder predictions – only one method, IUPRED, was used – amino acid composition, collection of DB annotations. However, the manuscript can be interesting for those interested in more realistic proteome analyses that include further variables – for example, temperature.

The realistic proteome analyses mentioned are something that is planned as a next step after this pilot study.

Line 24: “specific domains” might become “specific regions”.

This has been addressed.

Lines 57-58: The sentence “wet lab … tools” should be rewritten. It its present form, it makes no sense.

This sentence has been reworked to be clearer.

Section 2.1: perhaps one might write how many larvae have been used and how many of them were exposed to low temperature.

We are unsure if this would be redundant, since it was already stated how many larvae were used per experimental group.

Line 237: perhaps “self-made” might become “home-made”.

This has been addressed.

Lines 481-482: the sentence “In silico structure predictions … structural studies,” does not make sense and should be rewritten.

The sentence has been updated to make a distinction between what is done in silico and in vitro.

At this regard, the Authors should consider alphaFold2 and the related database developed at the EMBL.

The database created by predicting structures of the proteins with AlphaFold2 contains a comprehensive list of human proteins, which prevents its direct use for the structure predictions of the proteins in the current study. While predicting the structures of each protein would theoretically be possible, it is not covered in the scope of this study, and we do not have access to sufficient computational recourses to complete the task within a reasonable timeframe. On the other hand, while AlphaFold2 is indeed a highly precise disorder predictor, IUPred provides similarly reliable predictions regarding the disorder content of a protein. Since no accurate structure is generated by AlphaFold2 for disordered segments (disorder is annotated by the increase in the unreliability of the structure prediction), using it would not result in relevant additional information.

Reviewer 3 Report

The authors developed an approach that combines experimental and bioinformatical methods to identify and analyze intrinsically disordered proteins (IDPs) in cold hardy insect species. The approach involves IDP enrichment by sample heating and double-digestion of proteins, followed by peptide and protein identification. Then, proteins are bioinformatically analyzed for disorder content, presence of long disordered regions, amino acid composition and processes they are involved in. The work is interesting and requires a lot of effort and time. I have several naïve questions:

  1. 3 ‘One microtube from each sample was placed in a water bath at 100 °C for 5 minutes in order to remove globular proteins and enrich the content of IDPs’.

What is a mechanism of such enrichment? Dissolution of multi-subunit complexes, which are invisible for LC-MS/MS (p. 6)? At the same time, some proteins can aggregate at high temperatures, forming large complexes, which are also invisible for LC-MS/MS. Why the authors think that only globular proteins can precipitate at high temperatures? Some IDPs can also precipitate at high temperatures…

What was the purpose of digestion of the samples by trypsin? Trypsin could digest the IDP proteins and this could make it difficult to identify them in the resulting polypeptide mixture.

What proteins have been identified? The list of the proteins could be presented in a Supplement. Where from were taken their amino acid sequences?

What proteins contained long IDRs?

The authors worked with two types of samples: obtained from non-cold acclimated diapausing group; and from cold-acclimated diapausing group. They compared the content of ADPs in these two types of samples. At the same time, in Discussion, they have not discussed in detail these results, which would be interesting.

Author Response

The authors would like to thank the Reviewer for considering the article and hope that their questions have been adequately addressed. Please find below the response to the review:

The authors developed an approach that combines experimental and bioinformatical methods to identify and analyze intrinsically disordered proteins (IDPs) in cold hardy insect species. The approach involves IDP enrichment by sample heating and double-digestion of proteins, followed by peptide and protein identification. Then, proteins are bioinformatically analyzed for disorder content, presence of long disordered regions, amino acid composition and processes they are involved in. The work is interesting and requires a lot of effort and time. I have several naïve questions:

‘One microtube from each sample was placed in a water bath at 100 °C for 5 minutes in order to remove globular proteins and enrich the content of IDPs’.

What is a mechanism of such enrichment? Dissolution of multi-subunit complexes, which are invisible for LC-MS/MS (p. 6)? At the same time, some proteins can aggregate at high temperatures, forming large complexes, which are also invisible for LC-MS/MS. Why the authors think that only globular proteins can precipitate at high temperatures? Some IDPs can also precipitate at high temperatures… 

We did not intend to imply that IDPs do not precipitate at high temperatures, only that globular proteins are more susceptible to sample heating. This is supported by the several articles which we have cited, that use heating as a means of IDP content enrichment. The denatured globular proteins (and heat-susceptible IDPs) are removed by centrifugation and the remaining supernatant has a higher IDP content than the samples which were not heated (as can be seen from the presented Results).

What was the purpose of digestion of the samples by trypsin? Trypsin could digest the IDP proteins and this could make it difficult to identify them in the resulting polypeptide mixture. 

Trypsin digestion is a common step when preparing the samples for LC-MS/MS analysis and to prepare peptides that will be used for protein identification. It is only after identification that proteins are analyzed for disorder content and IDPs are identified.

What proteins have been identified? The list of the proteins could be presented in a Supplement. Where from were taken their amino acid sequences? 

What proteins contained long IDRs? 

Supplementary material has been added that contains the identified proteins as well as the information on their disorder content and long IDRs. The amino acid sequences were taken from the NCBI database.

The authors worked with two types of samples: obtained from non-cold acclimated diapausing group; and from cold-acclimated diapausing group. They compared the content of ADPs in these two types of samples. At the same time, in Discussion, they have not discussed in detail these results, which would be interesting.

The present article was a pilot study and was planned to be more of a technical paper to report on the methodology which could be used going forward. The IDP content between the two sample types was compared only to show how this particular experimental design leads to proteome changes in general, we did not intend to go into deeper ecophysiological analysis in this paper.

Round 2

Reviewer 1 Report

The authors have essentially modified the paper according to the reviewers suggestions and in my opinion the paper can be published.

Reviewer 3 Report

I am satisfied with the authors' answers and the changes they made to the manuscript.